# PERCEPTION-AWARE POINT-BASED VALUE ITERATION FOR PARTIALLY OBSERVABLE MARKOV DECISION PROCESSES

## ABSTRACT

Partially observable Markov decision processes (POMDPs) are a widely-used framework to model decision-making with uncertainty about the environment and under stochastic outcome. In conventional POMDP models, the observations that the agent receives originate from fixed known distribution. However, in a variety of real-world scenarios the agent has an active role in its perception by selecting which observations to receive. Due to combinatorial nature of such selection process, it is computationally intractable to integrate the perception decision with the planning decision. To prevent such expansion of the action space, we propose a greedy strategy for observation selection that aims to minimize the uncertainty in state. We develop a novel point-based value iteration algorithm that incorporates the greedy strategy to achieve near-optimal uncertainty reduction for sampled belief points. This in turn enables the solver to efficiently approximate the reachable subspace of belief simplex by essentially separating computations related to perception from planning. Lastly, we implement the proposed solver and demonstrate its performance and computational advantage in a range of robotic scenarios where the robot simultaneously performs active perception and planning.

## 1 INTRODUCTION

In the era of information explosion it is crucial to develop decision-making platforms that are able to judiciously extract useful information to accomplish a defined task. The importance of mining useful data appears in many applications including artificial intelligence, robotics, networked systems and Internet of things. Generally in these applications, a decision-maker, called an agent, must exploit the available information to compute an optimal strategy toward a given objective.

Partially observable Markov decision processes (POMDPs) provide a framework to model sequential decision-making with partial perception of the environment and under stochastic outcomes. The flexibility of POMDPs and their variants in modeling real-world problems has led to extensive research on developing efficient algorithms for finding near-optimal policies. Nevertheless, the majority of previous work on POMDPs either deal with sole perception or sole planning.

While independent treatment of perception and planning deteriorates performance, an integrated approach usually becomes computationally intractable. Thereupon, one must establish a trade-off between optimality and tractability when determining how much perception and planning rely on each other. We show that by restricting the perception to the class of subset selection problems and exploiting submodular optimization techniques, it is possible to partially decouple computing perception and planning policies while considering their mutual effect on the overall policy value.

In this work, we consider joint perception and planning in POMDPs. More specifically, we consider an agent that decides about two sets of actions; perception actions and planning actions. The perception actions, such as employing a sensor, only affect the belief of the agent regarding the state of the environment. The planning actions, such as choosing navigation direction, are the ones that affect the transition of the environment from one state to another. In subset selection problems, at each time step, due to power, processing capability, and cost constraints, the agent can pick a subset of available information sources along a planning action. The subset selection problem arise in various applications in control systems and signal processing, in wireless sensor networks, as well

as machine learning (Hashemi et al., 2018) and have been widely-studied (Shamaiah et al., 2010; Krause & Guestrin, 2007). However, the previous work on sensor selection problems assume that the planning strategy is known, while in this work, we simultaneously learn a selection strategy and a planning strategy.

## 1.1 RELATED WORK

Exact POMDP solvers optimize the value function over all reachable belief points. However, finding exact solution to POMDPs is PSPACE-complete (Papadimitriou & Tsitsiklis, 1987) which deems solving even small POMDPs computationally intractable. This has led to extensive search for near-optimal algorithms. A common technique is to sample a finite set of belief points that approximate the reachable subspace of belief and apply value iteration over this set, e.g., (Sondik, 1978; Cheng, 1988; Lovejoy, 1991; Zhang & Zhang, 2001; Spaan & Vlassis, 2005; Pineau et al., 2006). Pineau et al. (2006) proved that the errors due to belief sampling is bounded where the bound depends on the density of the belief set. A well-established offline POMDP solver is SARSOP (Kurniawati et al., 2008). SARSOP, similar to HSVI (Smith & Simmons, 2012), aims to minimize the gap between the lower and upper bounds on the value function by guiding the sampling toward the belief points that are reachable under union of optimal policies. In this paper, we show that the proposed greedy observation selection scheme leads to belief points that are on expectation close to the ones from the optimal (with respect to uncertainty reduction) set of observations, and hence value loss is small.

An instance of active perception is dynamic sensor selection. Kreucher et al. (2005) proposes a reinforcement learning approach that uses Rènyi divergence to compute utility of sensing actions. Joshi & Boyd (2009) formulated a single step sensor selection problem as semi-definite programming, however, it lacks theoretical guarantee. In Kalman filtering setting, Shamaiah et al. (2010) developed a greedy selection scheme with near-optimal guarantee to minimize log-determinant of the error covariance matrix of estimated state. Some prior work such as (Spaan, 2008; Spaan & Lima, 2009; Natarajan et al., 2015) model active perception as a POMDP. However, the most relevant work to ours are that of (Araya et al., 2010; Spaan et al., 2015; Satsangi et al., 2018). Araya et al. (2010) proposed $\rho$POMDP framework where the reward depends on entropy of the belief. Spaan et al. (2015) introduced POMDP-IR where the reward depends on accurate prediction about the state. Satsangi et al. (2018) established an equivalence property between $\rho$POMDP and POMDP-IR. Furthermore, they employed the submodularity of value function, under some conditions, to use greedy scheme for sensor selection. The main difference of our work is that we consider active perception as a means to accomplishing the original task while in these work, the active perception is the task itself and hence the POMDP rewards are metrics to capture perception quality.

The problem of selecting an optimal set of sensors from a ground set under cardinality constraint is NP-hard (Williams & Young, 2007). This hardness result has motivated design of greedy algorithms since they make polynomial oracle calls to the objective function. Additionally, if the objective function is monotone non-decreasing and submodular, Nemhauser et al. (1978) showed that a greedy selection achieves $(1 - 1/e)$ approximation factor. Mirzasoleiman et al. (2015) and Hashemi et al. (2018) developed randomized greedy schemes that accelerate the selection process for monotone submodular and weak-submodular objective functions, respectively. Krause & Guestrin (2007); Krause & Golovin (2014) have introduced different submodular information-theoretic objectives for greedy selection and have studied the theoretical guarantees of their maximization under different constraints. Here, we use the entropy of belief to capture the level of uncertainty in state and aim to select a subset of sensors that leads to maximum expected reduction in entropy. We employ the monotonicity and submodularity of the proposed objective to establish near-optimal approximation factor for entropy minimization.

## 1.2 CONTRIBUTIONS

A summary of our contributions are as follows:

- *Formulating the active perception problem for POMDPs:* We introduce a new mathematical definition of POMDPs, called AP$^2$-POMDP, that captures active perception as well as planning. The objective is to find deterministic belief-based policies for perception and planning such that the expected discounted cumulative reward is maximized.

- *Developing a perception-aware point-based value iteration algorithm:* To solve $AP^2$-POMDP, we develop a novel point-based method that approximates the value function using a finite set of belief points. Each belief point is associated with a perception action and a planning action. We use the near-optimal guarantees for greedy maximization of monotone submodular functions to compute the perception action while the planning action is the result of Bellman optimality equation. We further prove that greedy perception action leads to an expected reward that is close to that of optimal perception action.

## 2 PROBLEM FORMULATION

This section starts by giving an overview of the related concepts and then stating the problem.

### 2.1 PRELIMINARIES

The standard POMDP definition models does not capture the actions related to perception. We present a different definition which we call $AP^2$-POMDP as it models active perception actions as well as original planning actions. The active perception actions determine which subset of sensors (observations) the agent should receive. We restrict the set of states, actions, and observations to be discrete and finite.

#### 2.1.1 POMDP WITH PERCEPTION ACTION

We formally define an $AP^2$-POMDP below.

**Definition 1.** *An $AP^2$-POMDP is a tuple $\mathcal{P} = (S, A, k, T, \Omega, O, R, \gamma)$, where*

- *$S$ is the finite set of states.*

- *$A = A^{pl} \times A^{pr}$ is the finite set of actions with $A^{pl}$ being the set of planning actions and $A^{pr}$ being the set of perception actions. $A^{pr} = \{\delta \in \{0,1\}^n \mid |\delta|_0 \leq k\}$ constructs an $n$-dimensional lattice. Each component of an action $\delta \in A^{pr}$ determines whether the corresponding sensor is selected.*

- *$k$ is the maximum number of sensor to be selected.*

- *$T : S \times A^{pl} \times S \to [0, 1]$ is the probabilistic transition function.*

- *$\Omega = \Omega^1 \times \Omega^2 \times \ldots \times \Omega^n$ is the partitioned set of observations, where each $\Omega_i$ corresponds to the set of measurements observable by sensor $i$.*

- *$O : S \times A \times \Omega \to [0, 1]$ is the probabilistic observation function.*

- *$R : S \times A^{pl} \to \mathbb{R}$ is the reward function, and*

- *$\gamma \in [0, 1]$ is the discount factor.*

At each time step, the environment is in some state $s \in S$. The agent takes an action $\beta \in A^{pl}$ that causes a transition to a state $s' \in S$ with probability $Pr(s'|s, \beta) = T(s, \beta, s')$. At the same time step, the agent also picks $k$ sensors by $\delta \in A^{pr}$. Then it receives an observation $\omega \in \Omega$ with probability $Pr(\omega|s', \beta, \delta) = O(s', \beta, \delta, \omega)$, and a scalar reward $R(s, \beta)$.

**Assumption 1.** *We assume that the observations from sensors are mutually independent given the current state and the previous action, i.e., $\forall I_1, I_2 \subseteq \{1, 2, \ldots, n\}, I_1 \cap I_2 = \emptyset$ : $Pr(\bigcup_{i_1 \in I_1} \omega^{i_1}, \bigcup_{i_2 \in I_2} \omega^{i_2}|\mathbf{s}, \beta) = Pr(\bigcup_{i_1 \in I_1} \omega^{i_1}|\mathbf{s}, \beta) Pr(\bigcup_{i_2 \in I_2} \omega^{i_2}|\mathbf{s}, \beta)$.*

Let $\zeta(\delta) = \{i|\delta(i) = 1\}$ to denote the subset of sensors that are selected by $\delta$. If Assumption 1 holds, then:

$$Pr\left(\omega|s', \beta, \delta\right) = Pr\left(\bigcup_{i \in \zeta(\delta)} \omega^i|s', \beta, \delta\right) = \prod_{i \in \zeta(\delta)} O_i(s', \beta, \omega^i), \tag{1}$$

where $Pr(\omega^i|s', \beta) = O_i(s', \beta, \omega^i)$.

The belief of the agent at each time step, denoted by $b_t$ is the posterior probability distribution of states given the history of previous actions and observations, i.e., $h_t = (a_0, \omega_1, a_1, \ldots, a_{t-1}, \omega_t)$. A well-known fact is that due to Markovian property, a sufficient statistics to represent history of actions and observations is belief (Åström, 1965; Smallwood & Sondik, 1973). Given the initial belief $b_0$, the following update equation holds between previous belief $b$ and the belief $b_b^{',a,\omega}$ after taking action $a = (\beta, \delta)$ and receiving observation $\omega$:

$$
\begin{aligned}
b_b^{',a,\omega}(s') &= \frac{Pr\left(\omega|s',\beta,\delta\right)\sum_s Pr(s'|s,\beta)b(s)}{Pr\left(\omega|\beta,\delta\right)} \\
&= \frac{\prod_{i\in\zeta(\delta)} O_i(s',\beta,\omega^i)\sum_s T(s,\beta,s')b(s)}{\sum_{s'}\prod_{i\in\zeta(\delta)} O_i(s',\beta,\omega^i)\sum_s T(s,\beta,s')b(s)}.
\end{aligned}
\tag{2}
$$

The goal is to learn a deterministic policy to maximize $\mathbb{E}[\sum_{t=0}^{\infty} \gamma^t R(s_t, \beta_t)|b_0]$. A deterministic policy is a mapping from belief to actions $\pi : B \to A$, where $B$ is the set of belief states. Note that $B$ constructs a $(|S|-1)$-dimensional probability simplex.

The POMDP solvers apply value iteration (Sondik, 1978), a dynamic programming technique, to find the optimal policy. Let $V$ be a value function that maps beliefs to values in $\mathbb{R}$. The following recursive expression holds for $V$:

$$
V_t(b) = \max_a \left( \sum_{s\in S} b(s)R(s,a) + \gamma \sum_{\omega\in\Omega} Pr(\omega|b,a)V_{t-1}(b_b^{',a,\omega}) \right).
\tag{3}
$$

The value iteration converges to the optimal value function $V^*$ which satisfies the Bellman's optimality equation (Bellman, 1957). Once the optimal value function is learned, an optimal policy can be derived. An important outcome of (3) is that at any horizon, the value function is piecewise-linear and convex (PWLC) (Smallwood & Sondik, 1973) and hence, can be represented by a finite set of hyperplanes. Each hyperplane is associated with an action. Let $\alpha$'s to denote the corresponding vectors of the hyperplanes and let $\Gamma_t$ to be the set of $\alpha$ vectors at horizon $t$. Then,

$$
V_t(b) = \max_{\alpha\in\Gamma_t} \alpha.b.
\tag{4}
$$

This fact has motivated approximate point based solvers that try to approximate the value function by updating the hyperplanes over a finite set of belief points.

### 2.1.2 SUBMODULARITY

Since the proposed algorithm is founded upon the theoretical results from the field of submodular optimization, here, we overview the necessary definitions. Let $\mathcal{X}$ to denote a ground set and $f$ a set function that maps an input set to a real number.

**Definition 2.** *Set function* $f : 2^{\mathcal{X}} \to \mathbb{R}$ *is monotone nondecreasing if* $f(T_1) \le f(T_2)$ *for all* $T_1 \subseteq T_2 \subseteq \mathcal{X}$.

**Definition 3.** *Set function* $f : 2^{\mathcal{X}} \to \mathbb{R}$ *is submodular if*

$$
f(T_1 \cup \{i\}) - f(T_1) \ge f(T_2 \cup \{i\}) - f(T_2)
$$

*for all subsets* $T_1 \subseteq T_2 \subset \mathcal{X}$ *and* $i \in \mathcal{X}\backslash T_2$. *The term* $f_i(T_1) = f(T_1\cup\{i\})-f(T_1)$ *is the marginal value of adding element* $i$ *to set* $T_1$.

Monotonicity states that adding elements to a set increases the function value while submodularity refers to diminishing returns property.

### 2.2 PROBLEM DEFINITION

Having stated the required background, next, we state the problem.

**Problem 1.** *Consider a AP²-POMDP* $\mathcal{P} = (S, A, k, T, \Omega, O, R, \gamma)$ *and an initial belief* $b_0$. *We aim to learn a policy* $\pi(b) = (\beta, \delta)$ *such that the expected discounted cumulative reward is maximized, i.e,*

$$
\pi^* = \operatorname*{argmax}_\pi \mathbb{E}[\sum_{t=0}^{\infty} \gamma^t R(s_t, \pi(b_t))|b_0].
\tag{5}
$$

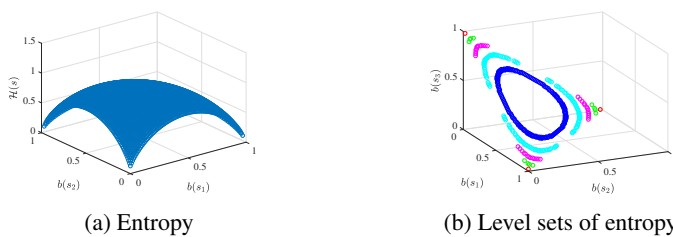

(a) Entropy          (b) Level sets of entropy

Figure 1: Entropy of belief for a 3-state POMDP.

It is worth noting that the perception actions affect the belief and subsequently the received reward in the objective function.

## 3   ACTIVE PERCEPTION WITH GREEDY SCHEME

For variety of performance metrics, finding an optimal subset of sensors poses a computationally challenging combinatorial optimization problem that is NP-hard. Augmenting POMDP planning actions with $\binom{n}{k}$ active perception actions results in a combinatorial expansion of the action space. Thereupon, it is infeasible to directly apply existing POMDP solvers to Problem 1. Instead of concatenating both sets of actions and treating them similarly, we propose a greedy strategy for selecting perception actions that aims to pick the sensors that result in minimal uncertainty about the state. The key enabling factor is that the perception actions does not affect the transition, consequently, we can decompose the single-step belief update in (2) into two steps:

$$\tilde{b}_b^{\beta}(s') = \sum_{s'} T(s, a, s')\tilde{b}(s), \tag{6a}$$

$$b_{\tilde{b}}^{'\delta,\omega}(s'') = \frac{\prod_{i\in\zeta(\delta)} O_i(s', \beta, \omega^i)\tilde{b}(s')}{\sum_{s'}\prod_{i\in\zeta(\delta)} O_i(s', \beta, \omega^i)\tilde{b}(s')}. \tag{6b}$$

This in turn implies that after a transition is made, the agent should pick a subset of observations that lead to minimal uncertainty in $b_{\tilde{b}}^{'\delta,\omega}$.

To quantify uncertainty in state, we use Shannon entropy of the belief. For a discrete random variable $x$, the entropy is defined as $\mathcal{H}(x) = -\sum_i p(x_i)\log p(x_i)$. An important property of entropy is its strict concavity on the simplex of belief points, denoted by $\Delta_B$ (Cover & Thomas, 2012). Further, the entropy is zero at the vertices of $\Delta_B$ and achieves its maximum, $\log|S|$, at the center of $\Delta_B$ that corresponds to uniform distribution, i.e., when the uncertainty about the state is the highest. Figure 1 demonstrates the entropy and its level sets for $|S| = 3$. Since the observation values are unknown before selecting the sensors, we optimize conditional entropy that yields the expected value of entropy over all possible observations. For discrete random variables $x$ and $y$, conditional entropy is defined as $\mathcal{H}(x|y) = \mathbb{E}_y[\mathcal{H}(x|y)] = \sum_i p(y_i)\mathcal{H}(x|y_i)$. Subsequently, with some algebraic manipulation, it can be shown that the conditional entropy of state given current belief with respect to $\delta$ is:

$$\mathcal{H}(\mathbf{s}|b, \delta) = - \sum_{\omega^{i_1}\in\Omega^{i_1}} \cdots \sum_{\omega^{i_k}\in\Omega^{i_k}} \sum_{s\in S} \left( b(s) \prod_{i_j\in\zeta(\delta)} O_{i_j}(s, \beta, \omega^{i_j}) \right.$$
$$\left. \log\left( \frac{b(s)\prod_{i_j\in\zeta(\delta)} O_{i_j}(s, \beta, \omega^{i_j})}{\sum_{s'\in S} b(s')\prod_{i_j\in\zeta(\delta)} O_{i_j}(s', \beta, \omega^{i_j})} \right) \right), \tag{7}$$

where $\zeta(\delta) = \{i_1, i_2, \ldots, i_k\}$. It is worth mentioning that $b$ is the current distribution of $\mathbf{s}$ and is explicitly written only for the purpose of better clarity, otherwise, $\mathcal{H}(\mathbf{s}|b, \delta) = \mathcal{H}(\mathbf{s}|\delta)$.

To minimize entropy, we define the objective function as the following set function:

$$f(\zeta) = \mathcal{H}(\mathbf{s}|\tilde{b}_b^{\beta}) - \mathcal{H}(\mathbf{s}|\tilde{b}_b^{\beta}, \bigcup_{i\in\zeta}\omega^i) \tag{8}$$

and the optimization problem as:

$$\delta^* = \arg\max_{\delta \in A^{pr}} f(\zeta(\delta)). \tag{9}$$

We propose a greedy algorithm, outlined in Algorithm 1 to find a near-optimal, yet efficient solution to (9). The algorithm takes as input the agent's belief and planning action. Then it iteratively adds elements from the ground set (set of all sensors) whose marginal gain with respect to $f$ is maximal and terminates when $k$ observations are selected.

---

**Algorithm 1** Greedy policy for perception action

---

1: **Input:** AP$^2$-POMDP $\mathcal{P} = (S, A, k, T, \Omega, O, R, \gamma)$, Current belief $b$, Planning action $\beta$.
2: **Output:** Perception action $\delta$.
3: Initialize $\mathcal{X} = \{1, 2, \ldots, n\}$, $\zeta = \emptyset$.
4: **for** $l = 1, \ldots, k$ **do**
5:     $j^* = \text{argmax}_{j \in \mathcal{X} \backslash \zeta} -\mathcal{H}(\mathbf{s}|\tilde{b}_b^\beta, \bigcup_{i \in \zeta \cup \{j\}} \boldsymbol{\omega}^i)$
6:     $\zeta \leftarrow \zeta \cup \{j^*\}$
7: **end for**
8: **return** $\delta$ corresponding to $\zeta$.

---

Next, we derive a theoretical guarantee for the performance of the proposed greedy algorithm. The following lemma states the required properties to prove the theorem. The proof of the lemma follows from monotonicity and submodularity of conditional entropy (Ko et al., 1995). See the appendix for the complete proof.

**Lemma 1.** *Let $\Omega = \{\boldsymbol{\omega}^1, \boldsymbol{\omega}^2, \ldots, \boldsymbol{\omega}^n\}$ to represent a set of observations of the state $\mathbf{s}$ that conditioned on the state, are mutually independent (Assumption 1 holds). Then, $f(\zeta)$, defined in (8), realizes the following properties:*

1. *$f(\emptyset) = 0$,*

2. *$f$ is monotone nondecreasing, and*

3. *$f$ is submodular.*

The above lemma enables us to establish the approximation factor using the classical analysis in (Nemhauser et al., 1978).

**Theorem 1.** *Let $\zeta^*$ to denote the optimal subset of observations with regard to objective function $f(\zeta)$, and $\zeta^g$ to denote the output of the greedy algorithm in Algorithm 1. Then, the following performance guarantee holds:*

$$\mathcal{H}(\mathbf{s}|\tilde{b}_b^\beta, \bigcup_{i \in \zeta^g} \boldsymbol{\omega}^i) \leq \frac{1}{e}\mathcal{H}(\mathbf{s}|\tilde{b}_b^\beta) + \left(1 - \frac{1}{e}\right)\mathcal{H}(\mathbf{s}|\tilde{b}_b^\beta, \bigcup_{i \in \zeta^*} \boldsymbol{\omega}^i). \tag{10}$$

**Remark 1.** *Intuitively, one can interpret the minimization of conditional entropy as pushing the agent's belief toward the boundary of the probability simplex $\Delta_B$. Due to convexity of POMDP value function on $\Delta_B$ (Sondik, 1978), this in turn implies that the agent is moving toward regions of belief space that have higher value.*

Although Theorem 1 proves that the entropy of the belief point achieved by the greedy algorithm is close to the entropy of the belief point from the optimal solution, the key question is whether the value of these points are close. We assess this question in the following and show that at each time step, on expectation, the value from greedy scheme is close to the value from optimal observation selection with regard to (9). To that end, we first show that the distance between the two belief points is upper-bounded. Thereafter, using a similar analysis as that of Pineau et al. (2006), we conclude that the difference between value function at these two points is upper-bounded.

**Theorem 2.** *Let the agent's current belief to be $b$ and its planning action to be $\beta$. Consider the optimization problem in (9), and let $\delta^*$ and $\delta^g$ to denote the optimal perception action and the perception action obtained by the greedy algorithm, respectively. It holds that:*

$$\mathbb{E}[\|b^g - b^*\|_1] \leq C_1,$$

*where $b^*$ and $b^g$ are the updated beliefs according to (6) and $C_1$ is a constant value.*

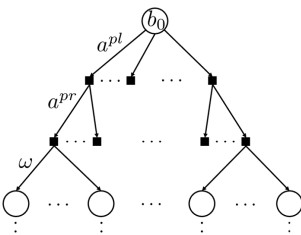

Figure 2: The belief reachability tree. The circles are belief points while squares depict branchings based on actions. Addition of perception actions leads to combinatorial expansion of number of belief points in each layer.

**Proof.** *We outline the sketch of the proof and bring the complete proof in the appendix. First, we show that minimizing conditional entropy of posterior belief is equivalent to maximizing Kullback-Leibler (KL-) divergence between current belief and the posterior belief, i.e., $D_{\mathcal{KL}}(b_b^{'\delta,\omega}\|b)$. Next, we exploit Pythagorean theorem for KL-divergence alongside its convexity to find a relation between $D_{\mathcal{KL}}(b_b^{'\delta^g,\omega}\|b)$ and $D_{\mathcal{KL}}(b_b^{'\delta^g,\omega}\|b_b^{'\delta^*,\omega})$ . Afterwards, using Pinkster's inequality, we prove that the total variation distance between $b_b^{'\delta^g,\omega}$ and $b_b^{'\delta^*,\omega}$ is bounded. This in turn yields the desired result on boundedness of $\|b^g - b^*\|_1$.*

**Theorem 3.** *Instate the notation and hypothesis of Theorem 2. Additionally, let $V$ to be the true value function for AP$^2$-POMDP. The following statement holds:*

$$\mathbb{E}[V(b^g) - V(b^*)] \leq C_2.$$

**Proof.** *The proof is omitted for brevity. See the appendix for the proof.*

## 4 PERCEPTION-AWARE POINT-BASED VALUE ITERATION

In this section, we propose a novel point-based value iteration algorithm to approximate the value function for AP$^2$-POMDPs. The algorithm relies on the performance guarantee of the proposed greedy observation selection in previous section. Before describing the new point-based solver, we first overview how point-based solvers operate. Algorithm 2 outlines the general procedure for a point-based solver. It starts with an initial set of belief points $B_0$ and their corresponding $\alpha$ vectors. Then it performs a Bellman backup for each point to update $\alpha$ vectors. Next, it prunes $\alpha$ vectors to remove dominated ones. Afterwards, it samples a new set of belief points and repeats these steps until convergence or other termination criteria is met. The difference between solvers is in how they apply sampling and pruning. The sampling step usually depends on the reachability tree of belief space, see Figure 2. The state-of-the-art point-based methods do not traverse the whole reachability tree, but they try to have enough sample points to provide a good coverage of the reachable space.

---

**Algorithm 2** Generic algorithm for point-based solvers (Araya et al., 2010)

---

1: **Input:** POMDP.
2: **Output:** Approximate value function $V$.
3: Initialize $B = B_0$ and $\Gamma_0$.
4: **while** $\sim$ (termination condition) **do**
5:     $B \leftarrow \text{Sample}(B)$
6:     $\Gamma \leftarrow \text{BackUp}(B, \Gamma)$
7:     $\Gamma \leftarrow \text{Prune}(B, \Gamma)$
8: **end while**
9: **return** $V(b) = \max_{\alpha \in \Gamma} \alpha.b$.

---

Note that the combinatorial number of actions due to observation selection highly expand the size of the reachability tree. To avoid dealing with perception actions in the reachability tree, we apply the greedy scheme to make the choice of $\delta$ deterministically dependent on $\beta$ and previous belief. To that end, we modify the BackUp step of point-based value iteration. The proposed BackUp step can be combined with any sampling and pruning method in other solvers, such as the ones developed by Spaan & Vlassis (2005); Kurniawati et al. (2008); Smith & Simmons (2012).

## 4.1 PROPOSED POINT-BASED SOLVER

In point-based solver each witness belief point is associated with an $\alpha$ vector and an action. Nevertheless, for AP$^2$-POMDPs, each witness point is associated with two actions, $\beta$ and $\delta$. We compute $\delta$ based on greedy maximization of (9) so that given $b$ and $\beta$, $\delta$ is uniquely determined. Henceforth, we can rewrite (3) using (4) to obtain:

$$
V_t(b) = \max_{(\beta,\delta)} \left( \sum_{s \in S} b(s) R(s,\beta) + \gamma \sum_{\omega \in \Omega} Pr(\omega|b,\beta,\delta) \max_{\alpha \in \Gamma_{t-1}} \alpha.b_b^{'\beta,\delta,\omega} \right)
$$

$$
= \max_{\beta} \left( \sum_{s \in S} b(s) R(s,\beta) + \right.
$$

$$
\left. \sum_{\substack{\omega \in \Omega_{i_1} \times \ldots \times \Omega_{i_k} \\ i_j \in \zeta(\bar{\delta})}} \max_{\alpha \in \Gamma_{t-1}} \sum_{s' \in S} \alpha(s') \times \prod_{i_j \in \zeta(\bar{\delta})} O_i(s',\beta,\omega^{i_j}) \sum_{s \in S} T(s,\beta,s') b(s) \right)
$$

$$
= \max_{\beta} \left( \sum_{s \in S} b(s) R(s,\beta) + \right.
$$

$$
\left. \sum_{\substack{\omega \in \Omega_{i_1} \times \ldots \times \Omega_{i_k} \\ i_j \in \zeta(\bar{\delta})}} \max_{\alpha \in \Gamma_{t-1}} \sum_{s \in S} \sum_{s' \in S} \alpha(s') \times \prod_{i_j \in \zeta(\bar{\delta})} O_i(s',\beta,\omega^{i_j}) T(s,\beta,s') b(s) \right).
$$

where $\bar{\delta} = \mathrm{argmax}_{\delta \in A^{pr}} f(\zeta(\delta))$ and $f$ is computed at $\tilde{b}_b^\beta$. This way, we can partially decouple the computation of perception action from the computation necessary for learning the planning policy.

Inspired by the results in the previous section, we propose the BackUp step detailed in Algorithm 3 to compute the new set of $\alpha$ vectors from the previous ones using Bellman backup operation. What distinguishes this algorithm from conventional Bellman backup step is the inclusion of perception actions. Basically, we need to compute the greedy perception action for each belief point and each action (Line 7). This in turn affects computation of $\Gamma_t^{b,\beta,\omega}$ as it represents a different set for each belief point (Lines 9-13). However, notice that this added complexity is significantly lower than concatenating the combinatorial perception actions with the planning actions and using conventional point-based solvers. See the appendix for detailed complexity analysis.

## 5 SIMULATION RESULTS

To evaluate the proposed algorithm for active perception and planning, we developed a point-based value iteration solver for AP$^2$-POMDPs. We initialized the belief set by uniform sampling from $\Delta_B$ (Devroye, 1986). To focus on the effect of perception, we did not apply a sampling step, i.e, the belief set is fixed throughout the iterations. However, one can integrate any sampling method such as the ones proposed by Smith & Simmons (2012); Kurniawati et al. (2008). The $\alpha$ vectors are initialized by $\frac{1}{1-\gamma} \min_{s,a} R(s,a).Ones(|S|)$ (Shani et al., 2013). Furthermore, to speedup the solver, one can employ a randomized backup step, as suggested by Spaan & Vlassis (2005). The solver terminates once the difference between value functions in two consecutive iterations falls below a predefined threshold. We also implemented a random perception policy that selects a subset of information sources, uniformly at random, at each backup step. We implemented the solver in Python 2.7 and ran the simulations on a laptop with 2.0 GHz Intel Core i7-4510U CPU and with 8.00 GB RAM.

## 5.1 ROBOTIC NAVIGATION IN 1-D GRID

The first scenario models a robot that is moving in a 1-D discrete environment. The robot can only move to adjacent cells and its navigation actions are $A^{pl} = \{left, right, stop\}$. The robot's transitions are probabilistic due to possible actuation errors. The robot does not have any sensor and it relies on a set of cameras for localization. There is one camera at each cell that outputs a probability

---

**Algorithm 3** BackUp step for AP$^2$-POMDP

---

1: **Input:** AP$^2$-POMDP $\mathcal{P} = (S, A, k, T, \Omega, O, R, \gamma)$, Current set of belief points $B_t$, Current set of $\alpha$ vectors $\Gamma_{t-1}$.
2: **Output:** Next set of $\alpha$ vectors $\Gamma_t$.
3: Initialize $\Gamma_t = \emptyset$, $\Gamma_t^{b,\beta} = \emptyset$ for all $b \in B_t$ and $\beta \in A^{pl}$.
4: **for** $\beta \in A^{pl}$ **do**
5:     $\Gamma_t^{\beta,*} \leftarrow \alpha^{\beta,*}(s) = R(s, \beta)$
6:     **for** $b \in B_t$ **do**
7:        $\bar{\delta} = \text{Greedy\_argmax}_{\delta \in A^{pr}} f(\zeta(\delta))$
8:        $\Gamma_t^{b,\beta,\omega} = \emptyset$
9:        **for** $\omega \in \Omega_{i_1} \times \ldots \times \Omega_{i_k}, i_j \in \zeta(\bar{\delta})$ **do**
10:          **for** $\alpha \in \Gamma_{t-1}$ **do**
11:            $\alpha^{b,\beta,\omega}(s) = \gamma \sum_{s' \in S} \prod_{i_j \in \zeta(\bar{\delta})} O_i(s', \beta, \omega^{i_j}) T(s, \beta, s') \alpha(s')$
12:            $\Gamma_t^{b,\beta,\omega} \leftarrow \Gamma_t^{b,\beta,\omega} \cup \alpha^{b,\beta,\omega}$
13:          **end for**
14:        **end for**
15:        $\alpha^{b,\beta} = \alpha^{\beta,*} + \sum_{\substack{\omega \in \Omega_{i_1} \times \ldots \times \Omega_{i_k} \\ i_j \in \zeta(\bar{\delta})}} \text{argmax}_{\alpha \in \Gamma_t^{b,\beta,\omega}} \alpha.b$
16:        $\Gamma_t^{b,\beta} \leftarrow \Gamma_t^{b,\beta} \cup \alpha^{b,\beta}$
17:     **end for**
18: **end for**
19: **for** $b \in B_t$ **do**
20:     $\alpha^b = \text{argmax}_{\alpha \in \Gamma_t^{b,\beta}, \beta \in A^{pl}} \alpha.b$
21:     $\Gamma_t = \Gamma_t \cup \alpha^b$
22: **end for**
23: **return** $\Gamma_t$

---

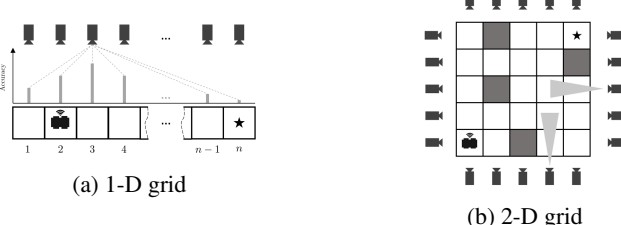

(a) 1-D grid

(b) 2-D grid

Figure 3: The robot moves in a grid while communicating with the cameras to localize itself. There is a camera at each state on the perimeter. The accuracy of measurements made by each camera depends on the distance of the camera from that state. The robot's objective is to reach the goal state, labeled by star, while avoiding the obstacles.

distribution over the position of the robot. The camera's certainty is higher when the robot's position is close to it. To model the effect of robot's position on the accuracy of measurements, we use a binomial distribution with its mean at the cell that camera is on. The binomial distribution represents the state-dependent accuracy. The robot's objective is to reach an specific cell in the map. For that purpose, at each time step, the robot picks a navigation action and selects $k$ camera from the set of $n$ cameras.

After the solver terminates, we evaluate the computed policy. To that end, we run 1000 iterations of Monte Carlo simulations. The initial state of the robot is the origin of the map and its initial belief is uniform over the map. Figure 4-(a) demonstrates the discounted cumulative reward, averaged over 1000 Monte Carlo runs, for random selection of 1 and 2 information sources, and greedy selection of 1 and 2 information sources. It can be seen that the greedy perception policy significantly outperforms the random perception. Figure 4-(b) depicts the belief entropy over the time. The lowest

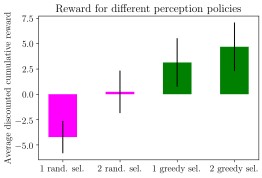

(a) Average discounted cumulative reward

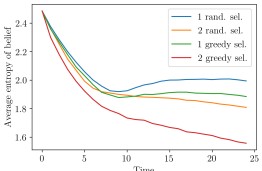

(b) Average entropy over time

Figure 4: Results of 1-D simulation for a map of size 12, averaged over 1000 runs for each perception policy. **Left:** The average discounted cumulative reward. The solid lines depict the corresponding standard deviations. **Right:** The average belief entropy over the time horizon of 25.

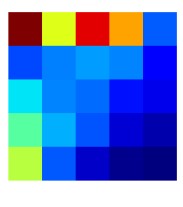
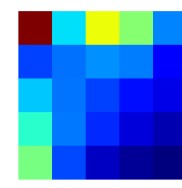
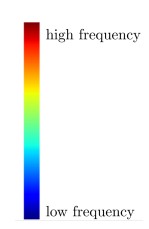

(a) Random perception algorithm

(b) Greedy perception algorithm

Figure 5: The frequency of visiting states when using different perception methods for a 2-D map of size 5*5 according to Figure 3-(b).

entropy of greedy perception, compared to random perception, shows less uncertainty of the robot when taking planning actions. See the appendix for further results.

## 5.2 ROBOTIC NAVIGATION IN 2-D GRID

The second setting is a variant of first scenario where the map is 2-D. Therefore the navigation actions of robot are $A^{pl} = \{up, right, down, left, stop\}$. The rest of the setting is similar to 1-D case, except the cameras' positions, as they are now placed around the perimeter of the map. Additionally, the robot has to now avoid the obstacles in the map. The reward is 10 at the goal state, -4 at the obstacles, and -1 in other states.

We applied the proposed point-based solver with both random perception and greedy perception on the 2-D example. Next, we let the robot to run for a horizon of 25 steps and we terminated the simulations once the robot reached the goal. Figure 5 illustrates the normalized frequency of visiting each state for each perception algorithm. It can be seen that the policy learned by greedy active perception leads to better obstacle avoidance. See the appendix for further results.

## 6 CONCLUSION

In this paper, we studied joint active perception and planning in POMDP models. To capture the structure of the problem, we introduced AP$^2$-POMDPs that have to pick a cardinality-constrained subset of observations, in addition to original planning action. To tackle the computational challenge of adding combinatorial actions, we proposed a greedy scheme for observation selection. The greedy scheme aims to minimize the conditional entropy of belief which is a metric of uncertainty about the state. We provided a theoretical analysis for the greedy algorithm that led to boundedness of value function difference between optimal entropy reduction and its greedy counterpart. Furthermore, founded upon the theoretical guarantee of greedy active perception, we developed a point-based value iteration solver for AP$^2$-POMDPs. The idea introduced in the solver to address active perception is general and can be applied on state-of-the-art point-based solvers. Lastly, we implemented and evaluated the proposed solver on a variety of robotic navigation scenarios.

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

## APPENDIX

In this section, we provide the proofs to the lemmas and theorems stated in the paper.

## PROOF OF THEOREM 1

First, in the next lemma, we show that the objective function defined for uncertainty reduction has the required properties for the analysis by Nemhauser et al. (1978), namely being normalized, monotone, and submodular.

**Lemma 1.** *Let $\Omega = \{\omega^1, \omega^2, \ldots, \omega^n\}$ to represent a set of observations of the state s that conditioned on the state, are mutually independent (Assumption 1 holds). Then, $f(\zeta)$, defined in (8), realizes the following properties:*

1. *$f(\emptyset) = 0$,*

2. *$f$ is monotone nondecreasing, and*

3. *$f$ is submodular.*

**Proof.** *Notice that $\tilde{b}_b^\beta$ is explicitly present to determine the current distribution of $\mathbf{s}$ and it is not a random variable. Therefore, for simplicity, we omit that in the following proof. It is clear that $f(\emptyset) = \mathcal{H}(\mathbf{s}|\tilde{b}_b^\beta) - \mathcal{H}(\mathbf{s}|\tilde{b}_b^\beta) = 0$.*

*Let $[n] = \{1, 2, \ldots, n\}$. To prove the monotonicity, consider $\zeta_1 \subset [n]$ and $j \in [n]\backslash\zeta_1$. Then,*

$$\mathcal{H}(\mathbf{s}| \bigcup_{i\in\zeta_1\cup\{j\}} \boldsymbol{\omega}^i)$$

$$\overset{(a)}{=} \mathcal{H}( \bigcup_{i\in\zeta_1\cup\{j\}} \boldsymbol{\omega}^i|\mathbf{s}) + \mathcal{H}(\mathbf{s}) - \mathcal{H}( \bigcup_{i\in\zeta_1\cup\{j\}} \boldsymbol{\omega}^i)$$

$$\overset{(b)}{=} \mathcal{H}( \bigcup_{i\in\zeta_1} \boldsymbol{\omega}^i|\mathbf{s}) + \mathcal{H}(\boldsymbol{\omega}^j|\mathbf{s}) + \mathcal{H}(\mathbf{s}) - \mathcal{H}( \bigcup_{i\in\zeta_1} \boldsymbol{\omega}^i)$$

$$- \mathcal{H}(\boldsymbol{\omega}^j| \bigcup_{i\in\zeta_1} \boldsymbol{\omega}^i)$$

$$\overset{(c)}{=} \mathcal{H}(\mathbf{s}| \bigcup_{i\in\zeta_1} \boldsymbol{\omega}^i) + \mathcal{H}(\boldsymbol{\omega}^j|\mathbf{s}) - \mathcal{H}(\boldsymbol{\omega}^j| \bigcup_{i\in\zeta_1} \boldsymbol{\omega}^i)$$

$$\overset{(d)}{=} \mathcal{H}(\mathbf{s}| \bigcup_{i\in\zeta_1} \boldsymbol{\omega}^i) + \mathcal{H}(\boldsymbol{\omega}^j|\mathbf{s}, \bigcup_{i\in\zeta_1} \boldsymbol{\omega}^i) - \mathcal{H}(\boldsymbol{\omega}^j| \bigcup_{i\in\zeta_1} \boldsymbol{\omega}^i)$$

$$\overset{(e)}{\leq} \mathcal{H}(\mathbf{s}| \bigcup_{i\in\zeta_1} \boldsymbol{\omega}^i) + \mathcal{H}(\boldsymbol{\omega}^j| \bigcup_{i\in\zeta_1} \boldsymbol{\omega}^i) - \mathcal{H}(\boldsymbol{\omega}^j| \bigcup_{i\in\zeta_1} \boldsymbol{\omega}^i)$$

$$= \mathcal{H}(\mathbf{s}| \bigcup_{i\in\zeta_1} \boldsymbol{\omega}^i),$$

*where $(a)$ and $(c)$ are due to Bayes' rule for entropy, $(b)$ follows from the conditional independence assumption and joint entropy definition, $(d)$ is due to the conditional independence assumption, and $(e)$ stems from the fact that conditioning does not increase entropy. The monotonicity of the objective function means that if the number of obtained observations are higher, the conditional entropy will be lower, and hence, on expectation, the uncertainty in the state will be lower.*

*Furthermore, from the third line of above proof, we can derive the marginal gain, i.e., the value of adding one sensor, as:*

$$f_j(\zeta_1) = \mathcal{H}(\mathbf{s}| \bigcup_{i\in\zeta_1} \boldsymbol{\omega}^i) - \mathcal{H}(\mathbf{s}| \bigcup_{i\in\zeta_1\cup\{j\}} \boldsymbol{\omega}^i)$$

$$= \mathcal{H}(\boldsymbol{\omega}^j| \bigcup_{i\in\zeta_1} \boldsymbol{\omega}^i) - \mathcal{H}(\boldsymbol{\omega}^j|\mathbf{s})$$

*To prove submodularity, let $\zeta_1 \subseteq \zeta_2 \subset [n]$ and $j \in [n]\backslash\zeta_2$. Then,*

$$f_j(\zeta_1) = \mathcal{H}(\boldsymbol{\omega}_j| \bigcup_{i\in\zeta_1} \boldsymbol{\omega}^i) - \mathcal{H}(\boldsymbol{\omega}_j|\mathbf{s})$$

$$\overset{(a)}{\geq} \mathcal{H}(\boldsymbol{\omega}_j| \bigcup_{i\in\zeta_1\cup(\zeta_2\backslash\zeta_1)} \boldsymbol{\omega}^i) - \mathcal{H}(\boldsymbol{\omega}_j|\mathbf{s})$$

$$\overset{(b)}{=} \mathcal{H}(\boldsymbol{\omega}_j| \bigcup_{i\in\zeta_2} \boldsymbol{\omega}^i) - \mathcal{H}(\boldsymbol{\omega}_j|\mathbf{s}) = f_j(\zeta_2),$$

*where $(a)$ is based on the fact that conditioning does not increase entropy, and $(b)$ results from $\zeta_1 \subseteq \zeta_2$. The submodularity (diminishing returns property) of objective function indicates that as the number of obtained observations increases, the value of adding a new observation will decrease.*

In the next theorem, we exploit the properties of the proposed objective function to analyze the performance of the greedy scheme.

**Theorem 1.** *Let $\zeta^*$ to denote the optimal subset of observations with regard to objective function $f(\zeta)$, and $\zeta^g$ to denote the output of the greedy algorithm in Algorithm 1. Then, the following performance guarantee holds:*

$$\mathcal{H}(\mathbf{s}|\tilde{b}_b^\beta, \bigcup_{i\in\zeta^g} \boldsymbol{\omega}^i) \leq \frac{1}{e}\mathcal{H}(\mathbf{s}|\tilde{b}_b^\beta) + \left(1 - \frac{1}{e}\right)\mathcal{H}(\mathbf{s}|\tilde{b}_b^\beta, \bigcup_{i\in\zeta^*} \boldsymbol{\omega}^i). \tag{11}$$

**Proof.** *The properties of $f$ stated in Lemma 1 along the theoretical analysis of greedy algorithm by Nemhauser et al. (1978) yields*

$$f(\zeta^g) \geq (1 - \frac{1}{e})f(\zeta^*).$$

*Using the definition of $f(\zeta)$ and rearranging the terms, we obtain the desired result.*

## PROOF OF THEOREM 2

Before stating the proof to Theorem 2, that bounds the distance of belief points from the greedy and optimal entropy minimization algorithms, we need to present a series of propositions and lemmas. Mutual information between two random variables is a positive and symmetric measure of their dependence and is defined as:

$$\mathcal{I}(\boldsymbol{x}; \boldsymbol{y}) = \sum_{x,y} p_{\boldsymbol{x},\boldsymbol{y}}(x, y) \log \frac{p_{\boldsymbol{x},\boldsymbol{y}}(x, y)}{p_{\boldsymbol{x}}(x)p_{\boldsymbol{y}}(y)}.$$

Mutual information, due to its monotonicity and submodularity, has inspired many subset selection algorithms (Krause & Golovin, 2014). In the following proposition, we express the relation between conditional entropy and mutual information.

**Proposition 1.** *Minimizing conditional entropy of the state with respect to a set of observations is equivalent to maximizing the mutual information of state and the set of observations. This equivalency is due to the definition of mutual information, i.e.,*

$$\mathcal{I}(\mathbf{s}; \bigcup_{i\in\zeta}\boldsymbol{\omega}^i) = \mathcal{H}(\mathbf{s}) - \mathcal{H}(\mathbf{s}|\bigcup_{i\in\zeta}\boldsymbol{\omega}^i), \tag{12}$$

*and the fact that $\mathcal{H}(\mathbf{s})$ is computed at $\tilde{b}_b^\beta$ which amounts to a constant value that does not affect selection procedure. Additionally, notice that (12) is the same as the definition of normalized objective function of greedy algorithm in (8).*

Another closely-related information-theoretic concept is Kullback-Leibler (KL-) divergence. The KL-divergence, also known as relative entropy, is a non-negative and non-symmetric measure of difference between two distributions. The KL-divergence from $q(x)$ to $p(x)$ is:

$$D_{\mathcal{KL}}(p_{\boldsymbol{x}}\|q_{\boldsymbol{x}}) = \sum_x p_{\boldsymbol{x}}(x) \log \left(\frac{p_{\boldsymbol{x}}(x)}{q_{\boldsymbol{x}}(x)}\right).$$

The following relation between mutual information and KL-divergence exists:

$$\begin{aligned}
\mathcal{I}(\boldsymbol{x}; \boldsymbol{y}) &= \sum_{x,y} p_{\boldsymbol{x},\boldsymbol{y}}(x, y) \log \frac{p_{\boldsymbol{x},\boldsymbol{y}}(x, y)}{p_{\boldsymbol{x}}(x)p_{\boldsymbol{y}}(y)} \\
&= \sum_y p_{\boldsymbol{y}}(y) \sum_x p_{\boldsymbol{x}|\boldsymbol{y}}(x|y) \log \frac{p_{\boldsymbol{x}|\boldsymbol{y}}(x|y)}{p_{\boldsymbol{x}}(x)} \\
&= \mathbb{E}_{\boldsymbol{y}}[D_{\mathcal{KL}}(p_{\boldsymbol{x}|\boldsymbol{y}}\|p_{\boldsymbol{x}})],
\end{aligned}$$

which allows us to state the next proposition.

**Proposition 2.** *The mutual information of state and a set of observations is the expected value of the KL-divergence from prior belief to posterior belief over all realizations of observations, i.e.,*

$$\mathcal{I}(\mathbf{s}; \bigcup_{i\in\zeta}\boldsymbol{\omega}^i) = \mathbb{E}_{\bigcup_{i\in\zeta}\boldsymbol{\omega}^i} \left[D_{\mathcal{KL}}(b_{\tilde{b}}^{',\delta,\omega}\|\tilde{b}_b^\beta)\right]. \tag{13}$$

$\tilde{b}_b^\beta$ *is the prior belief before selecting observations and $b_{\tilde{b}}^{',\delta,\omega}$ is the posterior belief after selecting perception action $\delta$ and receiving the observations.*

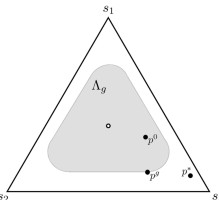

Figure 6: The probability simplex, $\Delta_B$, for a 3-state POMDP. Gray area illustrates the projection of hypograph of entropy, corresponding to posterior belief after greedy perception action, onto $\Delta_B$.

Let $p^0 := \tilde{b}_b^\beta$, $p^g := b_{\tilde{b}}^{'\delta,\omega}, \omega \sim \prod_{i \in \zeta^g} \Omega_i$, and $p^* := b_{\tilde{b}}^{'\delta,\omega}, \omega \sim \prod_{i \in \zeta^*} \Omega_i$ to denote prior belief (after taking planning action), posterior belief after greedy perception action, and posterior belief after optimal perception action, respectively. So far, we have established a relation between minimizing the conditional entropy of posterior belief and maximizing the expected KL-divergence from prior belief to posterior belief, i.e., $D_{\mathcal{KL}}(p^g \| p^0)$ (See Proposition 1 and Proposition 2). To relate $D_{\mathcal{KL}}(p^g \| p^0)$ and $D_{\mathcal{KL}}(p^g \| p^*)$, we state the next lemma. But first, we bring information-geometric definitions necessary for proving the lemma.

**Definition 4.** *Let $p$ to be a probability distribution over a finite alphabet. An I-sphere with center $p$ and radius $\rho$ is defined as:*

$$S(p, \rho) \stackrel{\text{def}}{=} \{q \mid D_{\mathcal{KL}}(q \| p) < \rho\}.$$

**Definition 5.** *Let $\Lambda$ to denote a convex set of probability distributions that intersect $S(p, \infty)$. A $\bar{q}$ satisfying*

$$D_{\mathcal{KL}}(\bar{q} \| p) \stackrel{\text{def}}{=} \min_{q \in \Lambda} D_{\mathcal{KL}}(q \| p),$$

*is called the I-projection of $p$ on $\Lambda$.*

**Lemma 2.** *Instate the definition of $p^0$, $p^g$, and $p^*$. The following inequality holds on expectation:*

$$D_{\mathcal{KL}}\left(p^0 \| p^*\right) \geq D_{\mathcal{KL}}\left(p^0 \| p^g\right) + D_{\mathcal{KL}}\left(p^g \| p^*\right).$$

**Proof.** *Consider the set $\Lambda_g = \{p \in \Delta_B | \mathcal{H}(p) \geq \mathcal{H}(p^g)\}$ that contain probability distributions whose entropy is lower-bounded by entropy of $p^g$. Since entropy is concave over $\Delta_B$, its hypographs are convex. Consequently $\Lambda_g$, the projection of a hypograph onto $\Delta_B$, is a convex set. Furthermore, due to monotonicity of conditional entropy, i.e., expected value of entropy over observations, we know that $p^0 \in \Lambda_g$. Besides, Due to optimality of $\zeta^*$, it holds that*

$$\mathcal{H}(\mathbf{s} | \tilde{b}_b^\beta, \bigcup_{i \in \zeta^*} \boldsymbol{\omega}^i) \leq \mathcal{H}(\mathbf{s} | \tilde{b}_b^\beta, \bigcup_{i \in \zeta^g} \boldsymbol{\omega}^i)$$

*which in turn yields $p^* \in \Delta_B \backslash \Lambda_g$. Figure 6 demonstrates these facts for an alphabet of size 3. $p^g$ is the I-projection of $p*$ on $\Lambda_g$. Therefore, by exploiting the analogue of Pythagoras' theorem for KL-divergence (Csiszár, 1975), we conclude:*

$$D_{\mathcal{KL}}\left(p^0 \| p^*\right) \geq D_{\mathcal{KL}}\left(p^0 \| p^g\right) + D_{\mathcal{KL}}\left(p^g \| p^*\right).$$

A direct result of the above lemma, after taking the expectation over $\bigcup_{i \in [n]} \boldsymbol{\omega}^i$, is:

$$\mathbb{E}_{\bigcup_{i \in \zeta^*} \boldsymbol{\omega}^i} \left[ D_{\mathcal{KL}}(\tilde{b}_b^\beta \| b_{\tilde{b}}^{'\delta,\omega}) \right] \geq \mathbb{E}_{\bigcup_{i \in \zeta^g} \boldsymbol{\omega}^i} \left[ D_{\mathcal{KL}}(\tilde{b}_b^\beta \| b_{\tilde{b}}^{'\delta,\omega}) \right] + \\ \mathbb{E}_{\bigcup_{i \in [n]} \boldsymbol{\omega}^i} \left[ D_{\mathcal{KL}}(b_{\tilde{b}}^{'\delta,\omega_{\zeta^g}} \| b_{\tilde{b}}^{'\delta,\omega_{\zeta^*}}) \right]. \tag{14}$$

In the following theorem, we use the stated lemma to bound the expected KL-divergence distance between greedy and optimal selection strategies.

**Theorem 4.** *The KL-divergence between $p^g$ and $p^*$ is upper-bounded, i.e.,*

$$\mathbb{E}_{\bigcup_{i \in [n]} \boldsymbol{\omega}^i} [D_{\mathcal{KL}}\left(p^g \| p^*\right)] \leq C_3,$$

*where $C_3$ is a constant.*

**Proof.** *Notice that while KL-divergence is not symmetric, the following fact still holds:*

$$\operatorname*{argmax}_{\zeta} \mathbb{E}_{\bigcup_{i\in\zeta} \boldsymbol{\omega}^i} \left[ D_{\mathcal{KL}}(b_{\tilde{b}}^{'\delta,\omega} \| \tilde{b}_b^{\beta}) \right] = \operatorname*{argmax}_{\zeta} \mathbb{E}_{\bigcup_{i\in\zeta} \boldsymbol{\omega}^i} \left[ D_{\mathcal{KL}}(\tilde{b}_b^{\beta} \| b_{\tilde{b}}^{'\delta,\omega}) \right],$$

*whenever the distributions consist of only non-zero elements. Once the algorithm is initialized with belief points that are not on the boundary of the probability simplex, this condition will hold. Also, notice that $\mathbb{E}_{\bigcup_{i\in\zeta^*} \boldsymbol{\omega}^i} \left[ D_{\mathcal{KL}}(\tilde{b}_b^{\beta} \| b_{\tilde{b}}^{'\delta,\omega}) \right]$ is constant. Now, Lemma 2 along the near-optimality result of greedy algorithm for entropy minimization (See Theorem 1) yield the desired result.*

Next, we bound the total variation distance between $p^g$ and $p^*$.

**Definition 6.** *The total variation distance between two probability distributions $p$ and $q$, over a countable state space $S$, is defined as:*

$$\mathrm{TVD}(p, q) = \frac{1}{2} \sum_{s \in S} |p(s) - q(s)|.$$

Pinsker's inequality bounds the total variation distance in terms of the KL-divergence (Cover & Thomas, 2012). Using this inequality, we can prove Theorem 2.

**Theorem 2.** *Let the agent's current belief to be $b$ and its planning action to be $\beta$. Consider the optimization problem in (9), and let $\delta^*$ and $\delta^g$ to denote the optimal perception action and the perception action obtained by the greedy algorithm, respectively. It holds that:*

$$\mathbb{E}[\|b^g - b^*\|_1] \le C_1,$$

*where $b^*$ and $b^g$ are the updated beliefs according to (6).*

**Proof.** *Note that the total variation distance is half of the $l_1$-norm. Hence,*

$$\mathbb{E}[\|b^g - b^*\|_1] = 2 \mathbb{E}[\mathrm{TVD}(p, q)]$$

$$\overset{(a)}{\le} 2 \mathbb{E}[\sqrt{\frac{1}{2} D_{\mathcal{KL}}(b^g \| b^*)}]$$

$$\overset{(b)}{\le} 2 \sqrt{\frac{1}{2} \mathbb{E}[D_{\mathcal{KL}}(b^g \| b^*)]}$$

$$\overset{(c)}{\le} \sqrt{2C_3},$$

*where $(a)$ is the result of applying Pinsker's inequality, $b$ is obtained by applying Jansen's inequality to concave square-root function, and $(c)$ follows from Theorem 4. The proof completes by taking $C_1 = \sqrt{2C_3}$.*

## PROOF OF THEOREM 3

In the next theorem, we use the bounded distance of belief points, obtained from the greedy and optimal entropy minimization algorithms to find an upper bound on the gap between the corresponding value functions.

**Theorem 3.** *Instate the notation and hypothesis of Theorem 2. Additionally, let $V$ to be the true value function for $AP^2$-POMDP. The following statement holds:*

$$\mathbb{E}[V(b^g) - V(b^*)] \le C_2.$$

**Proof.** *We use the PWLC property of value function. Let $\alpha^g$ and $\alpha^*$ to represent the gradient of value function at $b^g$ and $b^*$, respectively. Note that for every $\alpha$ vector, $\|\alpha\|_\infty \le \frac{\max\{|R_{max}|, |R_{min}|\}}{1-\gamma}$ where $R_{max} = \max_{s,\beta} R(s, \beta)$ and $R_{min} = \min_{s,\beta} R(s, \beta)$. Therefore, we can show that*

$$\mathbb{E}[V(b^g) - V(b^*)] = \mathbb{E}[\alpha^g.b^g - \alpha^*.b^*] = \mathbb{E}[\alpha^g.b^g - \alpha^g.b^* + \alpha^g.b^* - \alpha^*.b^*]$$

$$\overset{(a)}{\le} \mathbb{E}[\alpha^g.b^g - \alpha^g.b^* + \alpha^*.b^* - \alpha^*.b^*] = \mathbb{E}[\alpha^g.(b^g - b^*)]$$

$$\overset{(b)}{\le} \mathbb{E}[\|\alpha^g\|_\infty \|b^g - b^*\|_1]$$

$$\overset{(c)}{\le} C_1 \frac{\max\{|R_{max}|, |R_{min}|\}}{1 - \gamma},$$

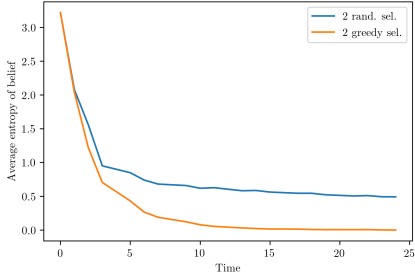

Figure 7: Comparing random and greedy selection for 2-D robotic navigation in terms of the average belief entropy over the time horizon of 25.

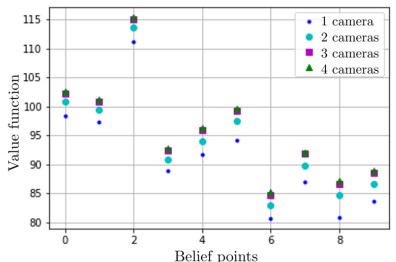

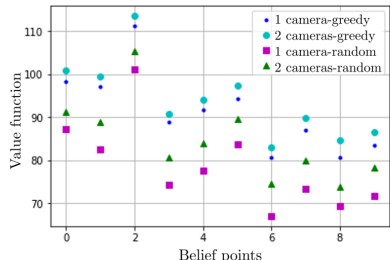

(a) Effect of varying number of selected cameras on the value function

(b) Effect of selection algorithm on the value function

Figure 8: The value function of a subset of size 10 from sampled belief points for 1-D navigation. The results show that the diminishing returns property of uncertainty reduction is propagated into value function.

*where $(a)$ follows from the fact that $\alpha^*$ is the gradient of the optimal value function, $(b)$ is due to Hölder's inequality, and $(c)$ is the result of Theorem 2. Taking $C_2 = C_1 \frac{\max\{|R_{max}|, |R_{min}|\}}{1-\gamma}$ yields the desired result.*

## COMPLEXITY ANALYSIS OF ALGORITHM 3

In this section, we compare the computational complexity of a point-based value iteration method that works with the concatenated action space, with the computational complexity of the proposed point-based method that picks the perception actions based on a greedy approach.

First, we compute the computations required for a single backup step in the point-based method with concatenated action space. To that end, consider a fixed set of sampled belief points $B$. Let $\Gamma$ to denote the current set of $\alpha$ vectors. Further, for the simplicity of analysis, assume that the number of possible observations from each information source is $|\Omega^i| = \bar{\Omega}, \forall i \in [n]$. The cardinality of a concatenated action space is $|A| = |A^{pr}||A^{pl}| = \binom{n}{k}|A^{pl}|$. Therefore, the complexity of a single backup step would be $\mathcal{O}(\binom{n}{k}|A^{pl}| \times \bar{\Omega}^n \times |\Gamma| \times |S|^2 + |B| \times \binom{n}{k}|A^{pl}| \times |S| \times \bar{\Omega}^n)$ (Shani et al., 2013).

On the other hand, applying greedy algorithm to pick a perception action requires $\mathcal{O}(n \times k)$ calls to an oracle that computes the objective function (or equivalently, the marginal gain). Here the objective function is the conditional entropy whose complexity with a naive approach in the $k^{\text{th}}$ iteration is $\mathcal{O}(\bar{\Omega}^k \times |S|^2)$. Therefore, applying Algorithm 3 as the backup step leads to $\mathcal{O}(|A^{pl}| \times |B| \times n \times k \times \bar{\Omega}^k \times |S|^2 + |A^{pl}| \times |B| \times \bar{\Omega}^k \times |\Gamma| \times |S|^2 + |B| \times |A^{pl}| \times |S| \times \bar{\Omega}^k)$ operations. Hence, the proposed approach, as a result of exploiting the structure of action space, would lead to significant computational gain, especially for large $n$.

ADDITIONAL NUMERICAL RESULTS

Figure 7 depicts the history of the belief entropy for the 2-D navigation when applying the proposed point-based method with random selection step and the proposed greedy selection step. As expected, the greedy selection leads to smaller entropy and hence, less uncertainty about the state. The corresponding average discounted cumulative reward after running 1000 Monte Carlo simulations is -18.8 for point-based value iteration with random selection step and -14.5 for point-based value iteration with greedy selection step, which demonstrates the superiority of the proposed method.

We further analyzed the effect of number of selected cameras on the agent's performance in the 1-D navigation scenario. Figure 8 illustrates the value function for a subset of sampled belief points after the algorithm has been terminated. It can be seen that the diminishing returns property of entropy with respect to number of selected observations is propagated through the value function as well.

