# OpenReview forum: "Perception-Aware Point-Based Value Iteration for Partially Observable Markov Decision Processes"
_ICLR.cc/2019/Conference_

### Official Review · AnonReviewer3 · 2018-11-01
**review for "Perception-Aware Point-Based Value Iteration for Partially Observable Markov Decision Processes"**

**Rating:** 7
**Confidence:** 2

**Review:**

Partially observable Markov decision processes (POMDPs) are a widely-used framework to model decision-making with uncertainty about the environment and under stochastic outcome. In conventional POMDP models, the observations that the agent receives originate from fixed known distribution. However, in a variety of real-world scenarios the agent has an active role in its perception by selecting which observations to receive. Due to the combinatorial nature of such a selection process, it is computationally intractable to integrate the perception decision with the planning decision.

The author proposes a new form of POMDPs called AP2-POMDP, which takes active perception into account. The AP2-POMDP problem restricts the maximum number of sensors that can be selected by an agent. The agent also faces the planning problem to select the sensors. To prevent such expansion of the action space, the authors propose a greedy strategy for observation selection and obtain a near optimal bound based on submodular optimization.

The author also proposes a greedy-based scheme for the agent to find an almost optimal active perception action by minimizing the uncertainty of beliefs. The author also uses theories to prove the near-optimal guarantees of this greedy method. The author also proposes a novel perception-aware point-based value iteration to calculate the value function and obtain the policy. The author also operates an interesting simulation experiment, which shows less uncertainty of the robot when taking planning actions when using the proposed solver.

The contribution is significant in reinforcement community. The writing is in general clear. It can be improved with minor modifications, for example, explaining math equations better in English.

My main comment for the authors is whether they have considered the scenario where the perception and the planning actions are connected. I agree with the authors that the best strategy for perception is to reduce uncertainty (and indeed, the greedy approach yields a near-optimal performance), given the restricted situation that the perception and planning are two separated processes. Nonetheless, in most real-world applications, the two processes are coupled, and therefore, we face, immediately, the trade-off between exploration and exploitation. I wonder if the authors have considered how they can extend their approach to such scenarios.

A few minor comments:


(i) The authors should add a legend and perhaps, more explanation in the captions of Figure 5. The colors of the heat-map are confusing. If dark blue and dark red represent lowest and highest frequency, what about other colors? Are there obstacles placed in the grid? If so, are they placed as shown in Figure 3(b)?

(ii) What is the effect of k, the maximum number of sensors to be placed? Can the authors provide a figure showing the change of performance with varying k?

(iii) It will be more convincing if the author deploys this algorithm to real-world robots and demonstrate its effectiveness.

---

> ### Author Response · Authors · 2018-11-27
> **Response to Reviewer 3**
>
> We thank the reviewer for the thoughtful and constructive feedback. Please find the authors’ response below.
>
> -----Explanation of theoretical parts
>
> *revision* The authors added more explanation to the theoretical proofs.
>
> -----Dependency of perception and planning actions
>
> The point mentioned by the reviewer is a very nice extension, for instance, considering the scenarios where the availability of information sources depends on the position of the agent. Adding such dependency makes the problem much harder, since, the belief points are almost-surly a combination of all states.
>
> However, this is one of the directions that the authors are currently following. One idea is to account for possible future uncertainty reduction with respect to each belief point and considering a branch-and-bound type algorithm, prune the planning actions that cannot obtain desired uncertainty given current level of uncertainty. This would lead to a heuristics-based method that limits the planning actions considered at each belief point.
>
> -----Caption of Figure 5
>
> *revision* We revised the caption and added the color spectrum.
>
> -----Effect of cardinality constraint
>
> ‘k’ is a given integer, coming from the physical constraint in the problem, not a parameter from the algorithm. However, increasing ‘k’ leads to less uncertainty as more measurements are obtained, due to monotonicity of entropy. Furthermore, this reduction has diminishing-returns property, due to submodularity of entropy.
>
> *revision* In a new appendix, the authors illustrate the effect of ‘k’ on the value function for each sampled belief point.
>
> -----Application over real-world robots
>
> As part of the future work, the authors plan to extend the simulation to a swarm of UAVs, following individual tracking tasks while communicating their sensory data with a few selected UAVs in their communication range.

---

### Official Review · AnonReviewer2 · 2018-11-01
**Decomposition of observation acquisition and action planning in POMDPs:  Insufficient motivation and results**

**Rating:** 4
**Confidence:** 4

**Review:**

This work addresses the problem of decomposing the observation acquisition from action planning in POMDPs.  Unfortunately, the paper has two major weaknesses.  First it is hindered by a confusing motivation, and the lack of clarity on the real purpose of the work is a problem throughout. Second the experiments are insufficient given current standards in the literature.

1. Motivation:  The introduction suggests that the main motivation is to reduce the computational cost of treating all observations within planning (“one must establish a trade-off between optimality and tractability”), though later, different reasons are offered (“power, processing capability, and cost constraints”).  It seems that each of these poses different constraints, and depending on which we are most concerned with, a different approximation scheme should be selected. For example, if the real concern is tractability of the POMDP solution, I don’t see why it’s not possible to just acquire all the sensor information, and afterwards decide how to approximate the tracking (this by the way is what most point-based POMDP methods effectively do).  For the proposed AP^2-POMDP, possibly a more reasonable motivation is high cost of observation acquisition; a clean argument would have to be made about the class of problems for which this constraint is crucial, why cardinality of sensor is the right way to articulate this constraint.

2. Experimental results:  The domains selected for the experiments are too simple, given current standards in the literature.  Looking at the 1D and 2D domains is fine to illustrate specific properties of your methods.  But it does not support the claim that the proposed model is more scalable than standard POMDPs.  In such simple domains, why not include results for a point-based method?  They should work in the 1D domain, probably also in the 2D domain.  Also, the setting for the 1-D domain, with a camera in every cell, seems very artificial.  First, if sensors are expensive, why put a camera in every cell?  And if they are not expensive, then why do we need to reason about which sensor to use at each step?  And why just read from k cameras at every step?  These questions point back to the concern regarding what is the real motivation for this work.  For the 2D domain, there are not even quantitative results on cumulative reward.  To be convincing, the results would need to be on substantially more complex domains; there are several POMDP benchmarks that could be considered, e.g. those in the work of Kurniawati et al.

Other comments:
-	Assumption 1 states that the observations from sensors are mutually independent give the state and action.  Can you explain why this is reasonable?  Or whether this is a strong assumption (unlikely to be met in practice)?
-	Some of the bounds seem like they could be very loose in practice, even (in the worse case) worse than the default bound of (R_max-R_min)/(1-\gamma). For example in Thm 3, in the case where the L1 distance between the 2 beliefs is 2, this is worse than the default bound.  Did you check what is the bound for the domains in the experiments?  Is it tighter than this?
-	A key statement is on p.8: “this added complexity is significantly lower than concatening the combinatorial perception actions with the planning actions”.   It is important to support this statement, ideally with both a precise complexity analysis, and with empirical results showing the lesser performance of standard point-based methods.

Minor comment:
-	The referencing style is broken and should be fixed, in particular proper use of Author (year) in the text.
-	The derivations in the top part of p.4 (Eqn 2-4 & surrounding text) are confusing, given that these apply to a standard POMDP, whereas on the previous page your present the AP^2-POMDP model. It might be better in Sec.2 to first (briefly) introduce POMDPs, with Eqn 2-4, then introduce AP^2-POMDP in Sec.3.
-	P.5: “It is worth noting that the objective function does not explicitly depend on perception actions”. This is a confusing statement; V depends on observations through b_t.  The next sentence clarifies this, but it would be better to avoid the confusing statement.
-	Alg.2:  Add a reference beside the title (unless you claim it is new).  Maybe Pineau et al. 2003.
-	P.7: “can be combined with any sampling and pruning method in other solvers” -> Add references for such sampling & pruning methods.

---

> ### Author Response · Authors · 2018-11-27
> **Response to Reviewer 2**
>
> We thank the reviewer for the thoughtful and constructive feedback. Please find the authors’ response below .
>
> -----Clarifying motivation
>
> The cost of measurement acquisition and processing in terms of power, communication, and processing computations define the physical constraints on the agent’s perception. This leads to the definition of problem 1 where we capture these constraints, similar to majority of sensor selection problems, by a cardinality constraint. As the reviewer correctly points out, some problems may call for different constraints. For instance, a knapsack constraint can define non-uniform cost over sensors, leading to 0.5 approximation instead of 0.667 when using greedy algorithms. For homogenous information source, as in the simulations, uniform cost is reasonable. For heterogenous information sources, a knapsack constraint is a better choice.
>
> Having defined the problem from physical constraints and demands, now the computational complexity arises from combining the selection actions with planning actions. This complexity has motivated the proposed solver and the sentence “one must establish a trade-off between optimality and tractability” refers to the defined problem (with cardinality constraint). Otherwise, if there is no limitation on the number of selected sensors, as the reviewer mentioned, one can use all the measurements, leading to less uncertainty and without the complexity of selection.
>
> *revision* The reviewer's assessment is completely valid regarding shortcomings in explaining the motivation. To resolve that, we edited the introduction of the paper to better convey the motivation and remove the ambiguity.
>
> -----Stronger empirical results
>
> The backup step in almost all the point-based solvers are the same while the sampling and pruning steps rely on efficient heuristics. Therefore, comparing the proposed solver that has a different back-up step and a conventional sampling and pruning would not result in the desired comparison. We intentionally avoided to use a specific sampling method in order to keep the solver general enough such that it can be combined with any sophisticated sampling (and/or pruning) method.
>
> The benchmarks in Kurniawati et al. cannot be used without significant modification since they lack perception actions. The designed experimental scenarios are based on Satsangi et al. (2018) and Spaan & Lima (2009) where POMDPs are used for active perception. The authors absolutely agree that more complex and more realistic simulations will better represent the importance of the proposed solver and plan to perform more empirical analysis as part of future work on a swarm of UAVs with tracking tasks and limited communication.
>
> *revision* The authors included the numerical values for 2-D navigation in a new appendix.
>
> -----Reason behind assumption 1
>
> Due to assumption 1, the sensors’ measurements only depend on the state and action, therefore eliminating the sensors effect on each other’s measurement, e.g., through noise from magnetic fields. This assumption is realistic in many practical settings, especially if they are not in small scales, e.g., micro size.
>
> -----Tightness of bounds
>
> The theoretical analysis for finding the bound follows a similar procedure as that of the classical analysis of point-based methods. The only difference is that the distance appearing between belief points is from the difference between greedy vs. optimal approach, not from the density of points. As part of future work, the authors aim to refine the bound for special classes of measurement models, e.g., Gaussian measurements with bounded variance.
>
> -----Supporting computational advantage
>
> *revision* To support the statement “this added complexity is significantly lower than concatenating the combinatorial perception actions with the planning actions”, the authors added an appendix detailing the complexity and its comparison with standard methods.
>
> -----Minor comments
>
> *revision* The manuscript is revised according to the minor comments.

---

### Official Review · AnonReviewer4 · 2018-11-20
**Algorithmic/theoretical development is sound, but assumptions are questionable.**

**Rating:** 6
**Confidence:** 4

**Review:**

This paper proposes a planning algorithm for a restricted class of POMDPs where the sensing decisions do not have any bearing on the hidden state evolution, or any material cost in terms of reward. A sensing decision consists of querying k out of n sensors which yield independent measurements of the hidden state. In this setting, the authors propose an optimization 2-stage optimization strategy, the first stage tries to find the optimal "planning" action in a point-based fashion, whereas the second aims to find the the sensor configuration that reduces the entropy of the post-update belief-state. The key observation is that the entropy minimization step is submodular and can be approximated greedily. This in turn translates to policy approximation bounds via information geometric arguments.

The positive: the paper is well written (save for some contained parts), the algorithm looks to be generally sound. Altogether the paper makes good points and is an interesting read.

The negative:
* first, I think there is some wide-spread misuse of the term "nearly optimal". When talking about near-optimality, this usually refers to finding a (controllably) bounded approximation to the optimal policy/value function. However, here this refers to the error relative to the approximate solution produced by the 2-stage procedure of minimizing entropy, then making a planning decision. It is not clear to me to begin with that this approach would produce bounded policies/value functions. As a counter example, consider a state space consisting of two state variables S1, S2 which evolve independently with additive reward  R(S1, a)  + R(S2, a) with  R(S1, a) !=0, while R(S2, a) = 0 for all actions a. Now, there could be a sensing configuration that collapses the uncertainty over S2 completely, but does nothing over S1, and a different one that give some small reduction of uncertainty  over S1 and nothing over S2. The former may outperform the latter to any degree in terms of belief state entropy, but it will not lead to an optimal policy, since that entropy reduction is not value directed. Unless I misunderstand something, in which case the authors should clarify.

* Second,  the particular assumptions in this paper are quite restrictive. This paper generally reads like a solution that was fit to a problem. This really hurts the story of the paper. It would be a vast improvement if the authors could find at least one plausible problem where there's a compelling case for this particular configuration of assumptions and try to evaluate how well they do on that problem relative to some reasonable baseline.

Remarks:
* The belief state notation used in this paper impacts undue suffering upon the reader. It comes in the form of expressions with multi-level sub/superscipts and accents such as: "b prime subscript b tilde superscript a superscipt pr comma omega". This is extremely hard to parse and possibly unnecessary, as b prime subscript b tilde and b tilde are the only configurations of accents and subscripts that appear. These could just be called alpha and beta and the rest is clear from the context.
* The claim in theorem 4, the argmax is the same under both directions of the KL divergence, is not obvious. It is definitely not true for minimization, otherwise the I-projection and the M-projection would coincide. This should be argued. Alternatively, this point can be alltogether skipped, since Pinsker's bound, which is the only place this is used, does not depend on the direction of KL.

Overall, this paper raises some nice points, but with these problems it is not a clear accept.

---

> ### Author Response · Authors · 2018-11-27
> **Response to Reviewer 4**
>
> We thank the reviewer for the thoughtful and constructive feedback. Please find the authors’ response below.
>
> -----Clarifying “near-optimality”
>
> The reviewer's point is indeed valid in that the near-optimality is with respect to entropy objective. As stated by the reviewer, the entropy, while widely used as an uncertainty metric, may not be the best measure in some settings. However, the intractability of combining perception and planning actions calls for introducing a separate uncertainty measure. As part of the future work, the authors plan to design task-oriented perception schemes that take into account the given task/rewards to shape a non-uniform (with respect to reward) measure of uncertainty.
>
> *revision* We revised the manuscript by emphasizing on the choice of entropy as one possible measure and clarifying what the near-optimality refers to.
>
> -----Restrictive assumptions
>
> As the first step toward joint perception and planning for POMDPs, we focus on a special case of perception where the perception action is defined as selecting a limited number of available information sources. While restrictive, this encompasses an important type of perception that has been extensively studied in many applications in control systems and signal processing, in wireless sensor networks, as well as machine learning (Krause & Guestrin, 2007).
>
> *revision* We revised the introduction to better explain the applications.
>
> -----Simplifying belief notation
>
> *revision* We simplified the notation used for representing belief to improve the readability.
>
> -----Is argmax the same under both directions of KL divergence?
>
> We thank the reviewer for mentioning this important point. The argmax would be the same under both directions if the distributions are restricted to have non-zero elements.  This would be the case if the belief points are not on the boundary of the probability simplex.
>
> *revision* We added the required clarification regarding the distributions in the proof.

---

### Meta-Review · Area_Chair1 · 2018-12-17
**Borderline paper: strong assumptions enable simplified approximate planning for restricted POMDPs**

**Confidence:** 5
**Recommendation:** Reject

**Metareview:**

This was a borderline paper and a very difficult decision to make.

The paper addresses a potentially interesting problem in approximate POMDP planning, based on simplifying assumptions that perception can be decoupled from action and that a set of sensors exhibits certain conditional independence structure.  As a result, a simple approach can be devised that incorporates a simple greedy perception method within a point-based value iteration scheme.

Unfortunately, the assumptions the paper makes are so strong and seemingly artificial to the extent that they appear reverse engineered to the use of a simple perception heuristic.  In principle, such a simplification might not be a problem if the resulting formulation captured practically important scenarios, but that was not convincingly achieved in this paper---indeed, another major limitation of the paper is its weak motivation.  In more detail, the proposed approach relies on decoupling of perception and action, which is a restrictive assumption that bypasses the core issue of exploration versus exploitation in POMDPS.  As model of active perception, the proposal is simplistic and somewhat artificial; the motivation for the particular cost model (cardinality of the sensor set) is particularly weak---a point that was not convincingly defended in the discussion.  Perhaps the biggest underlying weakness is the experimental evaluation, which is inadequate to support a claim that the proposed methods show meaningful advantages over state-of-the-art approaches in important scenarios.  A reviewer also raised legitimate questions about the strength of the theoretical analysis.

In the end, the reviewers did not disagree on any substantive technical matter, but nevertheless did disagree in their assessments of the significance of the contribution.  This is clearly a borderline paper, which on the positive side, was competently executed, but on the negative side, is pursuing an artificial scenario that enables a particularly simple algorithmic approach.

Despite the lack of consensus, a difficult decision has to be made nonetheless.  In the end, my judgement is that the paper is not yet strong enough for publication.  I would recommend the authors significantly strengthen the experimental evaluation to cover off at least two of the major shortcomings of the current paper: (1) The true utility of the proposed method needs to be better established against stronger baselines in more realistic scenarios.  (2) The relevance of the restrictive assumptions needs to be more convincingly established by providing concrete, realistic and more challenging case studies where the proposed techniques are still applicable.  The paper would also be improved if the theoretical analysis could be strengthened to better address the criticisms of Reviewer 4.